# Novel Rigidochromic and Anti-Kasha Dual Emission Fluorophores Based on D-π-A Dyads as the Promising Materials for Potential Applications Ranging from Optoelectronics and Optical Sensing to Biophotonics and Medicine

**DOI:** 10.3390/ijms24065818

**Published:** 2023-03-18

**Authors:** Svetlana A. Lermontova, Maxim V. Arsenyev, Anton V. Cherkasov, Georgy K. Fukin, Andrey V. Afanasyev, Andrey V. Yudintsev, Ilya S. Grigoryev, Elena Yu. Ladilina, Tatyana S. Lyubova, Natalia Yu. Shilyagina, Irina V. Balalaeva, Larisa G. Klapshina, Alexandr V. Piskunov

**Affiliations:** 1G.A. Razuvaev Institute of Organometallic Chemistry of RAS, 603950 Nizhny Novgorod, Russia; 2Institute of Applied Physics of RAS, 603950 Nizhny Novgorod, Russia; 3Biological Faculty, Lobachevsky State University of Nizhny Novgorod, 603950 Nizhny Novgorod, Russia

**Keywords:** aryltricyanoethylenes, cyanoarylporphyrazines, rigidochromism, anti-Kasha fluorescence, dual emission

## Abstract

Today we see an increasing demand for new fluorescent materials exhibiting various sensory abilities due to their broad applicability ranging from the construction of flexible devices to bioimaging. In this paper, we report on the new fluorescent pigments AntTCNE, PyrTCNE, and PerTCNE which consist of 3–5 fused aromatic rings substituted with tricyanoethylene fragments forming D-π-A diad. Our studies reveal that all three compounds exhibit pronounced rigidochromic properties, i.e., strong sensitivity of their fluorescence to the viscosity of the local environment. We also demonstrate that our new pigments belong to a very rare type of organic fluorophores which do not obey the well-known empirical Kasha’rule stating that photoluminescence transition always occurs from the lowest excited state of an emitting molecule. This rare spectral feature of our pigments is accompanied by an even rarer capability of spectrally and temporally well-resolved anti-Kasha dual emission (DE) from both higher and lowest electronic states in non-polar solvents. We show that among three new pigments, PerTCNE has significant potential as the medium-bandgap non-fullerene electron acceptor. Such materials are now highly demanded for indoor low-power electronics and portable devices for the Internet-of-Things. Additionally, we demonstrate that PyrTCNE has been successfully used as a structural unit in template assembling of the new cyanoarylporphyrazine framework with 4 D-π-A dyads framing this macrocycle (Pyr_4_CN_4_Pz). Similarly to its structural unit, Pyr_4_CN_4_Pz is also the anti-Kasha fluorophore, exhibiting intensive DE in viscous non-polar medium and polymer films, which strongly depends on the polarity of the local environment. Moreover, our studies showed high photodynamic activity of this new tetrapyrrole macrocycle which is combined with its unique sensory capacities (strong sensitivity of its fluorescent properties to the local environmental stimuli such as viscosity and polarity. Thus, Pyr_4_CN_4_Pz can be considered the first unique photosensitizer that potentially enables the real-time combination of photodynamic therapy and double-sensory approaches which is very important for modern biomedicine.

## 1. Introduction

Aryltricyanoethylenes (ATCEs) are promising new radical bridging ligands for the design of new molecule-based magnets with low Curie temperatures, so their extensive study is highly relevant [1,2,3,4].

We have recently discovered the new reaction of the cyclic tetramerization of ATCE resulting in the formation of the cyanoarylporphyrazine framework [5,6,7]. Unusual for template synthesis of tetrapyrroles, ATCE template assembly on a metal cation takes place under mild conditions and provides significant reaction yields at room temperature. Most of the porphyrazine macrocycles composed of ATCE structural units are highly effective sensitizers in photodynamic therapy (PDT) which is rather typical for tetrapyrroles [7,8,9,10,11,12]. Unexpectedly we observed a strong dependence of the fluorescence parameters of cyanoarylporphyrazine on the environmental viscosity. Such photophysical property of such fluorophores makes them plausible candidates for optical probes of intracellular viscosity which are very important for many biomedical applications [13,14,15].

We discovered that among the variety of novel ATCEs used for the design and synthesis of cyanoarylporphyrazines some pigments with polycyclic aromatic fragments demonstrate very interesting optical and electrochemical properties (Figure 1). The compounds with polycyclic aromatic fragments with an extensive π-system can be used as potential materials for organic electronics [16,17,18,19,20,21,22,23,24,25,26,27,28,29].

In this study, we present a new small molecule non-fullerene acceptor 2-(perylen-3-yl)ethene-1,1,2-tricarbonitrile (PerTCNE) consisting of the fused pentacyclic aromatic group (perylene) substituted with an effective electron-withdrawing tricyanoethylene fragment at the position 3 (Figure 1). We discovered that this simple structural combination provides unexpected spectral and electrochemical properties potentially enabling the production of new effective non-fullerene small molecule acceptors with very high electron affinity (LUMO–4.00 eV). In addition, we have found that PerTCNE is a medium-bandgap small molecule non-fullerene acceptor (SM NFA) (1.62 eV). To our best knowledge, this is a unique electronic property for monomeric perylene derivatives. Medium-bandgap materials possess band gaps of 1.8–1.9 eV corresponding to optical absorption cut-off at ca. 680 nm. It has been recently established that their highest performance is achieved under artificial indoor lighting with a power conversion efficiency of 50–60%, which makes them highly applicable in organic photovoltaics (OPV)s. There is a growing interest in using photovoltaic (PV) cells as ambient energy harvesters, which harvest indoor light to power low-power electronics. This is a promising tool for the Internet-of-Things, which can be used in portable IoT devices and smart environments (homes, offices, buildings, etc.) [30,31,32]. Thus, we assume that PerTCNE can be of great interest in developing effective PV-devices for indoor applications.

PerTCNE is also capable to form active ion-radicals under light excitation which makes it potentially applicable as the bipolarity enhanced photoinitiator for UV or two-photon 3D polymerization and microfabrication [33]. Indeed, we have found that PerTCNE has a high two-photon absorption cross-section which is close to that of the widely used photoinitiators for 3D polymerization.

To the great surprise, we have found that our small-molecule electron acceptors possess peculiar fluorescence properties, namely the hypsochromic shift of the Stokes shift, which is against the empirical Kasha’s rule. Moreover they possess dual emissions from both higher and lower electronic states it non-polar media. These findings have dramatically shifted our focus of interest toward these new pigments. According to recent investigations dual emission is very important for a variety of photonic applications, including sensing, optical memory, and bioimaging [34,35,36,37]. Considering this, we have addressed closer attention to PyrTCNE (2-(pyren-1-yl)ethene-1,1,2-tricarbonitrile) pigment. Interestingly, despite its sterically bulk pyrenyl substituents, it can be used as a structural unit for template assembling of the new unique cyanoarylporphyrazine (Pyr_4_CN_4_Pz), which shows excellent photodynamic properties and high sensitivity to local viscosity. We should note that Pyr_4_CN_4_Pz possesses an even more pronounced anti-Kasha fluorescence and dual emission effect, compared to PyrTCNE. This is due to the unique macrocycle framing consisting of dipolar D-π-A fragments throughout the perimeter of the macrocycle. The unique combination of photodynamic activity and double-sensory properties (rigidochromism and dual anti-Kasha emission) testifies to a high potential applicability of Pyr_4_CN_4_Pz in photonics, biophotonics, and medicine.

## 2. Results and Discussion

### 2.1. Synthesis and Characterization

The new ATCEs can be easily synthesized in 3 steps at room temperature (Figure 1). The general procedure for the synthesis of ArTCNE has been reported in the literature [1] as shown in Figure 1. Synthesis of **Pyr_4_CN_4_Pz** was carried out according to the scheme we described earlier in [11].

The first step is the reaction of an aromatic aldehyde with malonodinitrile in the presence of piperidine resulting in dicyanoethylene formation. The second step is the reaction of dicyanoethylene with KCN in the acidic environment giving corresponding aryl tricyanoethane. The third step is the dehydrogenation of the aryl tricyanoethane in the presence of chlorosuccinimide resulting in the formation of corresponding ATCE. The new compounds are highly colored substances soluble in most organic solvents (DMF, acetonitrile, THF, and toluene) and, to a lesser extent, in ether, chloroform, glycerol, and water.

### 2.2. Structural and Thermal Properties

The single crystal X-ray diffraction study showed that AntTCNE and PyrTCNE crystallize in monoclinic P21/c and P21/n space groups, respectively, with unique molecules of compounds in the asymmetric parts of unit cells. The molecular structures of AntTCNE and PyrTCNE are shown in Figure 2, and crystal data and structure refinement details are given in Appendix A. The crystal packings of both compounds do not contain additional solvent molecules. Tricyanoethylene groups in AntTCNE are disordered by two sites with occupancies of 0.61/0.39.

The C-N distances in tricyanoethylene fragments of AntTCNE and PyrTCNE range from 1.116(5) to 1.117(6) Å and from 1.141(4) to 1.144(5) Å, respectively, and are typical for this class of compounds. The C(2)-C(4) distances are 1.335(5), 1.336(5) Å in AntTCNE and 1.364(4) in PyrTCNE and correspond to the double C-C bonds [38] and are noticeably shorter than other C-C distances in these fragments (AntTCNE: 1.450(6)-1.454(7) Å; PyrTCNE: 1.432(5)-1.449(5) Å). TCNE groups in the studied compounds are rotated relatively to the polycyclic aromatic cores; the dihedral angles between the TCNE and Ant or Pyr planes are 64.0(2)°, 74.1(3)° (AntTCNE), and 45.18(4)° (PyrTCNE). The C(4)-C(6) distance in AntTCNE is 1.520(5), 1.521(5) Å and slightly longer that in PyrTCNE (1.474(4) Å).

The analysis of AntTCNE and PyrTCNE crystal packing showed that compounds are packed in stacks with the intermolecular π… π-stacking interactions between aromatic Ant or Pyr systems with resulting Ant…Ant and Pyr…Pyr distances 3.45(2)-3.48(2) Å and 3.48(2)-3.51(2) Å, respectively (for details see Appendix A). Crystal packing structures are poor in intermolecular hydrogen bonds (AntTCNE: N…H 2.66 Å; PyrTCNE: C…H 2.84-2.86 Å, H…H 2.33 Å). However, a detailed study of the intermolecular hydrogen interactions in these structures is problematic since the hydrogen atoms are placed and refined in calculated positions.

Unfortunately, the reflectivity of PerTCNE crystalline samples was insufficient to carry out an SC XRD study and establish its molecular and crystal structure. Therefore, we performed a theoretical study to compare the modeled molecular structure of PerTCNE with AntTCNE and PyrTCNE. DFT computations showed that the geometric characteristics of AntTCNE and PyrTCNE are fairly reproduced and compared to the experimental data (see Appendix A for details, Appendix A). The dihedral angles between TCNE and Ant, Pyr, or Per planes in the compounds are 65.9°, 51.1°, and 51.3° respectively, and this confirms an assumed non-planar molecular structure of PerTCNE.

The X-ray diffraction pattern of PerTCNE (Appendix A) is characterized by the clear reflexes indicated mainly by the microcrystal structure of the perylene-based compound. On the other hand, a slight baseline blur means the presence of a small amount of the amorphous phase in the sample. Despite that, high crystallinity can be beneficial for the formation of a favorable interpenetrating network with suitable phase separation contributing to higher electron mobility in the active layer of OPV [25,39].

No thermal effects indicating any chemical or phase transformations were observed in the temperature range of 23–270 °C (Appendix A). The melting of the compound was observed at 271 °C without any decomposition which was confirmed by the constant sample weight at the differential thermal analysis curve. A very slight decrease in sample weight in the temperature range 270–300 °C is presumably explained by partial evaporation of the liquid melt.

### 2.3. Photophysical Properties

AntTCNE, PyrTCNE, and PerTCNE are highly soluble in organic solvents such as DMSO, acetonitrile, and THF; less soluble in toluene, and the least soluble in glycerol and water. Molar extinction coefficients of AntTCNE, PyrTCNE, and PerTCNE in different solvents are given in Appendix A.

The normalized optical absorption spectra of AntTCNE, PyrTCNE, and PerTCNE in the acetonitrile solution (5 × 10^–6^ M) are shown in Figure 3.

The short-wavelength absorption bands which we observed for all three absorption spectra were attributed to electron transitions in the polycyclic cores of AntTCNE, PyrTCNE, and PerTCNE, although these bands demonstrate a much poorer vibronic structure compared to the corresponding individual aromatic compounds [40,41]. As for the long-wavelength part of the spectra presented in Figure 3, it was associated with the efficient charge transfer from the aromatic polycyclic core to the electron-withdrawing tricyanoethylene fragment. This band was the most red-shifted, and broadened for the perylene-containing compound PerTCNE.

As mentioned above, ATCEs can be used as the structural units in template assembly of the cyanoaryporphyrazine framework [7,8,9,10,11,12]. We discovered these macrocycles to be typical fluorescent molecular rotors demonstrating very strong rigidochromism, i.e., strong dependence on the fluorescent properties (emission intensity and fluorescent lifetime) on local viscosity. The physical principle of rigidochromism is the steric hindrance to intramolecular rotation, induced by Twisted Intramolecular Charge Transfer (TICT). TICT is formed upon photoexcitation in the strongly dipolar structures and is accompanied by twisting intramolecular motion in an excited state. The molecule enters a twisted state with lower excited-state energy, and relaxation from the twisted state is associated with either a red-shifted fluorescence emission or non-fluorescent relaxation [37]. Porphyrazines possess dipolar structural motifs D-π-A, where π is a π-conjugating bridge, D is an aromatic π-donor, and A is a tricarbonitrile acceptor, thus the mechanism of their rigidochromism can be attributed to TICT. Therefore, it is reasonable to expect that ATCEs including the same D-π-A motif as in porphyrazines can be noticeably sensitive to local viscosity. 

To compare the sensitivity to the viscosity of our compounds correctly, we should take into account that their fluorescence properties can be strongly affected by solvent polarity. So recording fluorescence spectra should be executed using solvents with approximately similar polarity and maximally different viscosity, e.g. in water (ε = 80, η = 1 cP) and glycerol (ε = 77, η = 950 cP) [15]. Indeed, the highest fluorescence intensity for PerTCNE, PyrTCNE, and AntTCNE is observed in highly viscous glycerol (Figure 4).

Unexpectedly, a controversial photophysical behavior of PyrTCNE was detected, namely a strong increase in fluorescence intensity in low-viscosity media (acetonitrile, DMSO, and water) upon the excitation at 350 nm. This could be explained by a combination of two possible photophysical mechanisms such as TICT and Association Induced emission (AIE) which, presumably, compete with each other under the excitation of vibronic structure (S_2_) levels belonging to the D-part of the D-π-A dyad. Intramolecular mobility induced by TICT may lead to strong conformeric heterogeneity [34]. Thus, the formation of some conformers which can associate and provide AIE effect should not be ruled out. In our opinion, the fluorescence enhancement in the polar low-viscosity media may be caused by the AIE mechanism arising from the strong conformeric heterogeneity. Interestingly, we also observe a strong increase in PerTCNE fluorescence intensity in the low-viscosity polar solvents upon the excitation at 400 nm (Appendix A). This means that in this case excitation of the lower level of PerTCN aromatic vibronic structure may increase the impact of the AIE effect into the interplay of two photophysical mechanisms. 

Both mentioned photophysical mechanisms (TICT and AIE) strongly contribute to Kasha’s rule suppression [40]. Further investigations of this complex interplay of TICT and AIE effects upon the excitation of aromatic vibronic structure should provide more deep insight into the photophysical behavior of the new anti-Kasha fluorophores.

Furthermore, the new pigments exhibit an additional important photophysical feature. They belong to the unique family of anti-Kasha fluorophores which exhibit dual emission (DE) from the lowest- and upper-lying states. Figure 5 presents the spectra of AntTCNE, PyrTCNE, and PerTCNE recorded in a non-polar environment (toluene). The DE behavior is considered important because it is a promising principle for various photonic and biophotonic applications [34].

As long as the TICT induces strong twisting of the molecular configuration, it is accompanied by a strong increase in the dipole moment in the twisted state [40]. We studied the transmission spectra of PerTCNE and the DMSO solution (10^−3^ mol/L) before and after UV irradiation for 20 min (405 nm, laser diodes, 30 and 120 J/cm^2^) and the dynamics of the optical density changes at a fixed wavelength (560 nm) for PerTCNE upon irradiation with doses of 30 and 120 J/cm^2^ (Figure 6a,b). We found that the curves of the light transmission changes under irradiation with laser power of 30 J/cm^2^ and 120 J/cm^2^ looked exactly the same. We applied an exponential decay approach to make sure that there was no noticeable difference in the dynamics of the optical density changes at a fixed wavelength (560 nm) for PerTCNE upon a fourfold increase in the laser radiation power, taking into account that the photo-induced breaking of the double bond in the tricyanoethylene fragment is the first-order reaction (Figure 6b and Appendix A). We established that difference in the dynamics of optical density with a fourfold increase in the laser radiation power is rather small. It means that the photoreaction rate weakly depends on the laser power, which confirms the resonance mechanism of the double bond break due to the TICT.

There was no change in the spectrum of unirradiated solution neither during this period (20 min) nor during a longer one (about 6 months). Thus, the double bond of PerTCNE which works, presumably, as π-spacer, conjugating condensed aromatic π-donor and electron acceptor fragment had been disrupted due to neo-UV irradiation inducing the powerful TICT. A gradual increase in light transmission up to the complete disappearance of the long-wavelength absorption band in the PerTCNE solution demonstrates the strong lability of the double bond during the charge transfer on the TICT mechanism under the influence of neo-UV radiation. The absorption related to the aromatic perylene fragment did not drop at all (Figure 6). Some reactive species, presumably, ion radicals, were formed as a result of the double bond disruption. As far as the TICT is regulated by environmental rigidity, the potential application of PerTCNE as a photoinitiator of viscosity-controlled polymerization could be developed.

Indeed, our primary experiments showed that these compounds can initiate photopolymerization of oligoether methacrylates and act as red-bleaching photoinitiators sensitive in the visible range. These data also confirm the generation of radical products during irradiation and will be studied in more detail in the future. Moreover, a three-dimensional microfabrication method based on two-photon polymerization with ultrashort laser pulses makes it very promising for PerTCNE to be used as a photoinitiator.

#### Two-Photon Absorption Investigations

Traditionally used z-scan method for studying two-photon absorption requires extremely high irradiation intensities, which destroy materials. We have optimized the approach, which allowed us to significantly reduce the intensity of laser radiation. This enabled us to store data and thus ensured higher setup sensitivity. When studying two-photon absorption of the new pigments, the method of the assignment of the number of photons arising during the relaxation of the excited state (fluorescence) as a result of two-photon absorption at a wavelength of 800 nm (the fundamental frequency of the laser) and one-photon absorption at a wavelength of 400 nm (the second harmonic of the laser) can be applied. Taking into account the fact that the photon emission during relaxation is a linear process with respect to the number of absorbed photons, this ratio allows us to measure the coefficient of two-photon absorption (*β*) [42]. When designing the equivalent configurations of laser beams at the fundamental frequency and at the second harmonic (the beams go coaxially near the surface of the cell), one can use the proportionality of the fluorescence intensity to the number of absorption events of two photons at the fundamental laser wavelength or a single photon at the second harmonic of the laser.

When the second harmonic is applied, the signal arrives at the power meter:(1)$$P^{400}=P_{0}{}^{400}\exp(-\alpha l),$$
where *l* is the cell length, and α is the linear absorption coefficient at a wavelength of 400 nm. In this case, energy had been absorbed:(2)$$P_{A}{}^{400}=P_{0}{}^{400}(1-\exp(-\alpha l))$$

The photon number and hence the number of the spectrometer counts for any of the fluorescence wavelengths (to improve the accuracy, we take the most intense spectral component *I*_*Fl*1_) are proportional to this absorbed energy:(3)$$P_{0}{}^{400}\exp(-\alpha l)=KI_{Fl2}$$

Otherwise, in terms of measured values:(4)$$P^{400}(\exp(\alpha l)-1)=KI_{Fl2}$$

A signal that arrives at the power meter when the fundamental frequency of the laser is applied, corresponds to
(5)$$P^{800}=P_{0}{}^{800}\exp(-\beta I^{800}l)-1)\approx P_{0}{}^{800}$$
where *β* is the coefficient of two-photon absorption, *I*^800^ is some average beam intensity which can be measured as
(6)$$I^{800}=\frac{P^{800}}{\pi r^{2}\tau f}$$
where *r* is the beam radius by level 1/e, *τ* (15 ns) is pulse duration, *f* is a pulse repetition rate, 10 Hz. In this case, the power value was absorbed in the medium with the two-photon absorption mechanism.
(7)$$P_{A}{}^{800}=P^{800}(\exp(\beta I^{800}l)-1)\approx P^{800}\beta I^{800}l$$

Since every two photons of the basic frequency (800 nm) provide the same energy impact as one photon of the second harmonic (400 nm), we can assume that with the same proportionality coefficient *K* which was used in (3).
(8)$$P^{800}\beta I^{800}l=KI_{Fl1}$$

Also, the spectral component energy of the fluorescence registered with an optical power meter (*I*_*Fl*2_) at the same wavelength and fluorescence energy registered with a spectrometer (*I*_*Fl*1_) are equal. However, the *I*_*Fl*2_ value is substantially less than *I*_*Fl*1_.

To avoid the hardware function embedded into coefficient K, we make the following proportion
(9)$$P^{800}\beta I^{800}lI_{Fl2}=I_{Fl1}P^{400}(\exp(\alpha l)-1)$$

It allows us to measure the two-photon absorption coefficient (*β*):(10)$$\beta =\frac{I_{Fl1}P^{400}(\exp(\alpha l)-1)}{P^{800}I^{800}lI_{Fl2}}=\frac{I_{Fl1}P^{400}(\exp(\alpha l)-1)\pi r^{2}\tau f}{{\left(P^{800}\right)}^{2}lI_{Fl2}}$$

To provide sufficient accuracy of the *β* value, the method applied here requires noticeable fluorescence intensity of the tested compound and **PerTCNE** had been selected for this experiment because it demonstrated the best compliance with this criterion. The value of *β* and two-photon absorption cross-sections (*δ*) for PerTCNE are presented in Table 1.

The value of the two-photon absorption coefficient *β* of PerTCNE is very close to that of Michler’s ketone (~2 × 10^−48^ cm^4^s/photon) and some similar compounds, which are widely used as two-photon polymerization initiators for three-dimensional optical data storage and microfabrication [33,43].

### 2.4. Electrochemical Investigation

The redox properties of AntTCNE, PyrTCNE, PerTCNE, and Pyr_4_CN_4_Pz assembled from PyrTCNE were studied by cyclic voltammetry in acetonitrile vs. Ag/AgCl/KCl(sat.). The cyclic voltammograms (CV) for both complexes exhibit closed current–voltage loops showing that the electrogenerated intermediates are stable in the time-frame of the experiment. The cyclic voltammograms are presented in Appendix A.

The results of electrochemical studies and the energies of molecular orbitals are presented in Table 2.

We observed the appropriate decrease of the bandgap of ATCEs (the energy difference between the HOMO and the LUMO) with the increase in the number of fused rings in the aromatic core. However, we did not expect to achieve such a small bandgap for the perylene-based compound, which puts it among the potential medium-bandgap acceptors (1.9–1.5 eV) for photovoltaic applications. As mentioned above, this opens up the prospect to apply PerTCNE as a small molecule non-fullerene electron acceptor for the fabrication of OPV active layers with an appropriate polymeric electron donor. The idea is particularly promising in terms of indoor photovoltaic devices since it was established that medium-bandgap materials allow reaching much higher OPV performance than that for narrow-bandgap materials (1.1–1.4 eV) under AM1.5G illumination because for indoor light harvesting only photons from of the visible light need to be collected [30].

As for PyrTCNE, a wide-bandgap electron acceptor (1.94 eV), it presumably can be used in combination with PerTCNE as the additive to increase the total light harvesting of visible light in the region of shorter wavelengths. The bandgap of porphyrazine Pyr_4_CN_4_Pz is very close to that of PyrTCNE but it is significantly wider than that of PerTCNE presumably due to the influence of the perylene fragment, which is known to show n-type, electron conducting behavior and electron-acceptor properties [44].

### 2.5. Template Assembling of the New Cyanoporphyrazine Frameworks Using AntTCNE, PyrTCNE and PerTCNE as the Structural Units

It was interesting to find out whether such spatially bulk fragments prevent the formation of a macrocyclic structure from AntTCNE, PyrTCNE, and PerTCNE. We found that despite a spatially hindered structure of PyrTCNE it easily forms a cyanoarylporphyrazine framework Pyr_4_CN_4_Pz as a result of template assembling of these molecules in the presence of bis(indenyl)ytterbium(II) (Figure 2). This synthesis was carried out according to the procedures we reported previously [15,45]:

However, this synthesis does not apply to AntTCNE and PerTCNE. The less spatially hindered anthracene aromatic core substituted by tricyanoethylene fragments at position 9 seems to create greater spatial obstacles to the assembly of the macrocyclic framework than the pyrrole aromatic core, while the 5-fused rings of perylene core are presumably just too bulky for porphyrazine macrocycle assembling. The absorption (a) and fluorescence (b) spectra of new porphyrazine Pyr_4_CN_4_Pz are presented in Figure 7.

The absorption spectrum of Pyr_4_CN_4_Pz presented in Figure 7 is rather typical for tetrapyrrole macrocycles, but its maximum is strongly moved to the longer wavelength in comparison with cyanoarylporphyrazines with peripheral monocyclic aromatic groups [15]. The fluorescence spectrum is unusually broadened towards the long wavelengths which can be explained by the partial association of macrocyclic rings in water. This opens up new avenues to use Pyr_4_CN_4_Pz in biomedical optical devices, since it provides better transparency of the biological tissue for light irradiation. In addition, we found that Pyr_4_CN_4_Pz demonstrates excellent photodynamic activity which is much higher than that we established for cyanoarylporphyrazines with monocyclic aryl framing [8,10,15]. Exposure of pretreated cell culture to light irradiation at the dose of 20 J/cm2 induced cell death. The half-maximal inhibitory concentration (IC_50_) for Pyr_4_CN_4_Pz was 1.0 μM in light conditions and above 150 μM when incubated in the dark (Figure 8).

The ratio of IC50 in the dark and under light irradiation is rather high (~150). It means that this photosensitizer is very toxic for irradiated cancer cells and not harmful for healthy tissues which are not irradiated. 

Light irradiation of Pyr_4_CN_4_Pz led to rapid degradation of singlet oxygen trap 1,3-diphenylisobenzofuran (DPBF) which supports the significant contribution of Photoreaction Type II in Pyr_4_CN_4_Pz phototoxicity. In a non-viscous DMSO solution (2 cP), the quantum yield of singlet oxygen generation was about 0.35 (Appendix A). However, as it was previously mentioned, the tested compound belongs to the fluorescent molecular rotors. The probability of formation of the particular states of porphyrazine molecule (excited singlet planar and excited singlet twisted, with one to four twisted groups, triplet planar and twisted) is strongly dependent on viscosity. In this regard, we expect an increase in the quantum yield of singlet oxygen generation in viscous environments (in particular, in cell membranes with a viscosity of about 800 cP) in which Pyr_4_CN_4_Pz is localized in the cell.

Despite the hindered aromatic framing of Pyr_4_CN_4_Pz similar to most cyanoporphyrazines, it retains a strong sensitivity of its fluorescent properties to the environmental viscosity due to the photoinduced intramolecular mobility of peripheral aromatic groups. [15]. It follows from the data in Figure 7c indicating a strong emission and viscosity increase. Therefore, Pyr_4_CN_4_Pz can be potentially used as an optical sensor of local viscosity. 

Similar to ATCEs, Pyr_4_CN_4_Pz is found to be a typical anti-Kasha fluorephore. Below we present the absorption and fluorescence spectra of Pyr_4_CN_4_Pz in an environment of different polarity and rigidity.

The fluorescence spectrum of Pyr_4_CN_4_Pz in toluene given below (Figure 9) is rather similar to that of PerTCNE (Figure 4). In a nonpolar medium, porphyrazine demonstrates spectrally and temporally resolved DE from the both lowest- and upper-lying states. Moreover, the intensity of emission from the lowest electronic state is very sensitive to local polarity. It drops with the environmental polarity increase. Such additional sensory capacity makes it possible to use rigidochromic Pyr_4_CN_4_Pz as a polarity sensor.

Moreover, in the media of various rigidity ranging from low-viscosity toluene to solid film of polycarbonatedimethacrylate (PCMA), the DE effect remains quite pronounced. However, the lowest wavelength emission maximum shifts significantly to shorter wavelengths in the media of high viscosity (Figure 9). Quantum yields of the PerTCN long-wavelength (LW) and short-wavelength (SW) DE emission of Pyr4CN4Pz in the different solvents are given in Table 3. Interestingly, the quantum yield ratio of long-wavelength (LW) and short-wavelength (SW) emissions strongly depends on solvent polarity and viscosity. We observed a strong anti-Kasha effect in a polar solvent with low viscosity. On the contrary, in a nonpolar solvent with high viscosity (castor oil), we observed a pronounced suppression of the anti-Kasha effect. But dual emission and a rather high fluorescence quantum yield at long wavelengths were found. Curiously, we observed an unexpectedly high quantum yield for the emissive S2 state (SW) relaxation at the upper-lying state in the aqueous medium (Table 3). This, presumably, means that in contrast to that at the low-lying state the AIE mechanism, significantly dominates here strengthening the anti-Kasha effect.

Interestingly, we detected high quantum yields of Pyr_4_CN_4_Pz in castor oil that is similar to cell membranes in terms of viscosity and polarity. We also observe a very strong DE effect in castor oil that is very promising for anticancer therapy monitoring through ratiometric evaluation of cell membrane polarity 

We proposed a new mechanism of DE for rigidochromic dyes such as Pyr_4_CN_4_Pz and PerTCNE in a non-polar low-viscosity medium based on the above spectral data (Figure 10).

Thus, we hypothesized a new photophysical mechanism of DE in our rigidochromic pigments: the rotated excited state resulting from the TICT can behave as a high-oscillator-strength (emissive) state while the S1 state is found to be a low-oscillator-strength (not emissive) state. A more rigid medium (such as castor oil and polymer film) strongly suppressed the TICT, but this effect does not disappear completely even in polymer film [37]. In this case, we observe a significantly reduced energy gap between the ground state and the rotated excited state compared to that in a nonpolar medium of low viscosity. This explains the hypsochromic shift of long-wavelength emission of Pyr_4_CN_4_Pz in the rigid environment. Nevertheless, the pronounced DE effect is also observed in the rigid media.

## 3. Materials and Methods

### 3.1. Reagents and Equipment

The reagents, for example, 9-anthracenecarboxaldehyde, 1-pyrenecarboxaldehyde, 3-perylenecarboxaldehyde, were used as received from Sigma-Aldrich or TCI without further purification. The ^1^H and ^13^C NMR spectra were recorded using a Brucker Avance DPX-200 spectrometer (200 MHz) and a Bruker Avance III 400 (100 MHz) respectively. All chemical shifts were referenced to the TMS lock signal. The UV–vis electron absorption spectra of the compounds were recorded using a Perkin Elmer Lambda 25 spectrometer. The IR spectra of the pigments in the form of mineral oil suspensions were recorded using an FSM 1201 spectrometer. Fluorescence was measured using a Synergy MX plate reader (BioTek, Winooski, VT, USA) in the range of 370–700 nm with excitation at a wavelength of 350 nm. Integration time at a single point—100 msec, slits width—9 nm, gain—100. The positive ion electron ionization mass spectra were measured using a PolarisQ/TraceGCUltra GC/MS spectrometer at 70 eV, at the temperature of ion source 230 °C, by heating the sample from 50 to 450 °C in the mass number range of 90–600.

The thermogravimetric analysis (TGA) and differential thermal analysis (DTA) were carried out using a DTG-60 H Simultaneous Thermogravimetric and Differential Thermal Analyzer (Shimadzu). PerTCNE powder (1 mg) was heated under flowing air at a heating rate of 5 °C per minute in the temperature range of 23–300 °C.

### 3.2. Powder X-ray Diffraction

The X-ray diffraction pattern was obtained using diffractometer Shimadzu Lab XRD-6000 (CuK**α**-filtered radiation, λ = 1.54178 Å) at 30 kV and 30 mA in the 2θ range of 10–50° at the rate of 4 degrees per minute.

### 3.3. Single Crystal X-ray Crystallography

The X-ray data for AntTCNE and PyrTCNE were collected using a Bruker D8 Quest diffractometer (MoKα-radiation, ω-scans technique, λ = 0.71073 Å, T = 298 K for AntTCNE and 100 K for PyrTCNE) using the APEX3 software package. The structures were solved via the intrinsic phasing algorithm and refined by full-matrix least squares against F2 using SHELX [37]. SADABS [36] was used to perform absorption corrections. All non-hydrogen atoms in AntTCNE and PyrTCNE were found from Fourier syntheses of electron density and refined anisotropically. All hydrogen atoms were placed in calculated positions and refined isotropically in the “riding” model with U(H)iso = 1.2 Ueq of their parent atoms. Both structures were refined as pseudo-merohedral twins with laws (1 0 0 0 −1 0 0 0 −1) for AntTCNE and (−1 0 0 0 −1 0 0.325 0 1) for PyrTCNE and resulting domain ratios 0.52/0.48 and 0.87/0.13, respectively.

The crystallographic data and structure refinement details are given in Appendix A. CCDC 2085342 (AntTCNE) and 2085343 (PyrTCNE) contain supplementary crystallographic data for this paper. These data are provided free of charge by The Cambridge Crystallographic Data Centre: ccdc.cam.ac.uk/structures (accessed on 20 November 2022). The corresponding CIF is also available as Appendix A.

### 3.4. Computational Details

The DFT calculations were performed using the Gaussian 09 software package [37] with the hybrid B3LYP functional [45,46] and aug-cc-DZVP basis set [46] for all atoms. The stationary points on the potential energy surfaces were located by full geometry optimization. The absence of imaginary frequencies suggests that the molecules are in the minimum of potential energy.

### 3.5. Synthetic Details

The general procedure for the synthesis of ArTCNE is reported in the literature [1] as shown in Figure 1.

#### 3.5.1. Synthesis of 1**a**–**c**

Malononitrile (7.3 mmol), carboxaldehyde (7.3 mmol) and a few drops of piperidine were dissolved in 150 mL of ethanol. The reaction mixture was stirred for 24 h at room temperature. The resulting precipitate was collected by filtration, rinsed thoroughly with ice-cold 95% ethanol, and dried under vacuum to 2-(aryl)-1,1-dicyanoethylene (1**a**–**c**).

2-(antracene-1-yl)-1,1-dicyanoethylene (**1a**). 78% yield. ^1^H NMR (200 MHz, CDCl_3_) δ (ppm): 8.95 (s, ^1^H), 8.65 (s, ^1^H), 8.10 (d, J = 8.92 Hz, 2H), 7.93 (d, J = 8.77 Hz, 2H), 7.72–7.54 (overlapped peaks, 4H). IR (KBr) (cm^−1^): 3055 (arC-H), 2230 (C≡N), 1622, 1578, 1551, 1520, 1486 (arC-C), 1561 (C=C).

2-(pyrene-1-yl)-1,1-dicyanoethylene (**1b**). 96% yield. ^1^H NMR (200 MHz, CDCl_3_) δ (ppm): 8.91 (s, ^1^H), 8.83 (d, J = 8.20 Hz, ^1^H), 8.39–8.10 (overlapped peaks, 8H). IR (KBr, cm^−1^): 3042 (arC-H), 2221 (C≡N), 1625, 1598, 1584, 1560, 1532, 1508, 1482 (arC-C, C=C). EI MS (70 eV): *m*/*z* (%) 278 (100) [(M)^+^].

2-(perylene-3-yl)-1,1-dicyanoethylene (**1c**). 74% yield. ^1^H NMR (200 MHz, CDCl_3_) δ (ppm): 8.50 (s, ^1^H), 8.36–8.24 (overlapped peaks, 5H), 7.87–7.79 (overlapped peaks, 3H), 7.71–7.57 (overlapped peaks, 3H). IR (KBr, cm^−1^): 3054 (arC-H), 2224 (C≡N), 1600, 1589, 1578, 1557, 1501 (arC-C, C=C). EI MS (70 eV): *m*/*z* (%) 328 (100) [(M)^+^].

#### 3.5.2. Synthesis of 2**a**–**c**

A solution of potassium cyanide (12.2 mmol) in cold water (50 mL) was added to 2-(aryl)-1,1-dicyanoethylene (6.1 mol) dissolved in ethanol (150 mL). Then, the reaction mixture was diluted with cold water (700 mL), stirred for 45 min at room temperature, and concentrated HCl was added (1.7 mL). After 15 min, the resulting mixture was refrigerated at 8 °C for 12 h. The precipitate was filtered off, carefully washed with water, and dried under vacuum to 2-(aryl)-1,1,2-tricyanoethane (**2a**–**c**).

2-(antracene-1-yl)-1,1,2-tricyanoethane (**2a**). 74% yield. ^1^H NMR (200 MHz, CDCl_3_) δ (ppm): 8.69 (s, 1H), 8.27 (d, J = 6.94 Hz, 2H), 8.15 (d, J = 8.48 Hz, 2H), 7.80–7.57 (overlapped peaks, 4H), 6.02 (d, J = 10.75 Hz, 2H), 4.95 (d, J = 10.75 Hz, 2H). IR (KBr, cm^−1^): 3056 (arC-H), 2258, 2245 (C≡N), 1626, 1597, 1530, 1486 (arC-C), 1557 (C=C).

2-(pyrene-1-yl)-1,1,2-tricyanoethane (**2b**). 70% yield. 1H NMR (200 MHz, CDCl_3_) δ (ppm): 8.40–8.01 (overlapped peaks, 9H), 5.60 (d, J = 5.96 Hz, ^1^H), 4.48 (d, J = 5.85 Hz, ^1^H). IR (KBr, cm^−1^): 3044 (arC-H), 2253 (C≡N), 1604, 1587, 1570, 1561, 1536, 1527, 1525, 1510, 1501 (arC-C, C=C). EI MS (70 eV): *m*/*z* (%) 305 (10) [(M)^+^].

2-(perylene-3-yl)-1,1,2-tricyanoethane (**2c**). 71% yield. ^1^H NMR (200 MHz, DMSO) δ (ppm): 8.51 (t, J = 8.22 Hz, 2H), 8.46–8.43 (overlapped peaks, 2H), 8.13 (d, J = 8.53 Hz, ^1^H), 8.46–8.43 (overlapped peaks, 3H), 7.71 (t, J = 8.01 Hz, ^1^H), 7.61–7.56 (overlapped peaks, 2H), 6.27 (d, J = 6.58 Hz, ^1^H), 5.93 (d, J = 6.56 Hz, ^1^H). IR (KBr, cm^−1^): 2263 (C≡N), 1600, 1589, 1567, 1520, 1501 (arC-C, C=C). EI MS (70 eV): *m*/*z* (%) 355 (12) [(M)^+^].

#### 3.5.3. Synthesis of AntTCNE, PyrTCNE, PerTCNE

Chlorosuccinimide (8 mmol) and 2-(aryl)-1,1,2-tricyanoethane (5.7 mmol) were dissolved in diethyl ether (100 mL). After 1 hour, 150 mL of water was added. The mixture was stirred until precipitation of a solid. The precipitate was filtered off and dried. The resulting product was chromatographically purified on a silica gel column eluting with THF to 2-(aryl)-1,1,2- tricyanoethene (ArTCNE).

2-(anthracene-1-yl)-1,1,2-tricyanoethene (AntTCNE). 65% yield. ^1^H NMR (200 MHz, CD3CN) δ (ppm): 8.85 (s, ^1^H), 8.18 (d, J = 8.13 Hz, 2H), 8.10 (d, J = 8.81 Hz, 2H), 7.78–7.62 (overlapped peaks, 4H). ^13^C NMR (100 MHz, CD3CN): 141.35, 133.52, 130.89, 129.55, 129.19, 126.44, 123.42, 120.03, 117.35, 113.96, 110.88, 110.06, 105.03. IR (KBr, cm^−1^): 3055 (arC-H), 2239 (C≡N), 1624, 1585, 1523, 1485 (arC-C), 1554 (C=C). EI MS (70 eV): *m*/*z* (%) 279 (100) [(M)^+^].

2-(pyrene-1-yl)-1,1,2-tricyanoethene (PyrTCNE). 92% yield. 1H NMR (200 MHz, DMSO) δ (ppm): 8.53–8.38 (overlapped peaks, 7H), 8.29 (d, J = 8.95 Hz, ^1^H), 8.21 (t, J = 7.67 Hz, 1H). ^13^C NMR (100 MHz, DMSO) δ (ppm): 140.58, 134.92, 131.28, 130.98, 130.91, 130.47, 128.96, 128.23, 128.15, 127.92, 127.58, 125.60, 124.07, 123.72, 123.41, 122.98, 115.39, 111.94. IR (KBr, cm^−1^): 3051 (arC-H), 2232 (C≡N), 1624, 1595, 1585, 1544, 1505, 1482 (arC-C, C=C). EI MS (70 eV): *m*/*z* (%) 303 (100) [(M)^+^].

2-(perylene-3-yl)-1,1,2-tricyanoethene (PerTCNE). 93% yield. ^1^H NMR (200 MHz, DMSO) δ (ppm): 8.48–8.37 (overlapped peaks, 4H), 8.00–7.96 (overlapped peaks, ^2^H), 7.89 (d, J = 8.12 Hz, ^1^H), 7.83 (d, J = 8.07 Hz, ^1^H), 7.75 (t, J = 8.12 Hz, ^1^H), 7.58–7.52 (overlapped peaks, 2H). ^13^C NMR (100 MHz, DMSO) δ (ppm): 139.49, 137.14, 134.28, 131.96, 131.67, 130.98, 130.92, 129.72, 129.56, 129.50, 129.15, 128.40, 127.63, 127.47, 125.81, 124.66, 123.99, 122.85, 122.60, 120.42, 115.12, 112.58, 111.95. IR (KBr, cm^−1^): 3054 (arC-H), 2234, 2224 (C≡N), 1599, 1589, 1567, 1545, 1535, 1520, 1498 (arC-C, C=C). EI MS (70 eV): *m*/*z* (%) 353 (100) [(M)+].

#### 3.5.4. Synthesis of 3,8,13,18-Tetra(pyrene-1-yl)-2,7,12,17-Tetracyanoporphyrazine (Pyr4CN4Pz)

PyrTCNE (0.62 g, 2.05 mmol) in THF (10 mL) and bis(indenyl)ytterbium (0.25 g, 0.46 mmol) in THF (2 mL) were mixed in an inert atmosphere. After 24 h, the solution was filtered under vacuum. To remove the unreacted compound PyrTCNE and its complex with ytterbium, the obtained solution was washed with degassed toluene until discoloration. Pyr_4_CN_4_PzYb was dried under reduced pressure and isolated (0.34 g, 0.24 mmol) 54%.

Pyr_4_CN_4_PzYb (0.28 g, 0.20 mmol) was dissolved in trifluoroacetic acid (2 mL) and stirred at room temperature for 30 min. The resulting solution was poured onto ice/water (30 mL) and a dark-blue solid was precipitated. This residue was centrifuged and carefully washed with water until the supernatant gave a neutral reaction. The product was further purified by chromatography (Silica gel 60, 40–63 µm particle size, Sigma-Aldrich, THF eluent). The purification was repeated twice and gave isolated Pyr_4_CN_4_Pz as a dark blue solid (0.06 g, 0.05 mmol, 25% yield). ^1^H NMR (200 MHz, CD3CN) δ (ppm): 7.10 (s, 1H), 7.35 (s, 1H), 8.35–7.68 (overlapped peaks, 36H). ^13^C NMR (400 MHz, DMSO-d6) δ (ppm): 163.93, 163.36,163.34, 163.26, 136.14, 136.11, 133.84, 133.44, 132.26, 131.24, 131.06, 130.81, 130.48, 129.82, 129.65, 129.20, 129.13, 128.97, 127.85, 127.72, 127.55, 127.52, 127.19, 126.79, 126.65, 126.49, 126.35, 126.29, 125.54, 125.03, 124.30, 124.25, 124.19, 124.00, 123.92, 123.64, 122.86, 122.78, 122.01, 121.58, 120.21, 120.14, 118.67, 118.62, 118.26, 117.29, 117.18, 68.72, 67.49, 66.18, 65.81, 65.79. IR (KBr, cm^−1^): 3413 (N-H), 3045 (arC-H), 2201 (C≡N), 1762 (C=N), 1596, 1584, 1498, 1480 (Car=Car). Anal. calcd. for C_84_H_38_N_12_: C 83.02, N 13.83%. Found (EDAX): C 82.85, N 13.94%.

### 3.6. Cell lines and Culturing Conditions

A431 human epidermoid carcinoma cell line was used (All-Russian Collection of Cell Cultures, Institute of Cytology of the Russian Academy of Sciences, Saint-Petersburg, Russia). A-431 cells were cultured in DMEM (PanEco, Russia), containing 10% of fetal bovine serum (HyClone, USA) and 2 mM of L-glutamine (PanEco, Russia) in a 5% CO_2_ atmosphere at 37 °C. For passaging the cells were detached with 0.25% trypsin-EDTA (PanEco, Russia).

### 3.7. Study of Photodynamic Activity of Pyr_4_CN_4_Pz In Vitro

*A431* cells were seeded in 96-well plates at a density of 4000 cells per well and were left to attach overnight (20 h). The medium was removed, and the cells were then incubated in the presence of Pyr_4_CN_4_Pz at different concentrations in a growth medium at 37 °C for 4 h in 5 % CO_2_. Then the cells were washed twice with PBS and the medium containing Pyr_4_CN_4_Pz was replaced by a growth medium. To estimate photoinduced toxicity, the cells were exposed to light irradiation (615–635 nm, 20 mW/cm^2^, 20 J/cm^2^) using a LED light source providing a homogeneous light distribution in 96-well plates [40]. Irradiated cells were then incubated for 24 h before cell viability was measured by the MTT assay. The growth medium was replaced with the solution of 3-(4,5-dimethyl-2-thiasolyl)-2,5-diphenyl-2H-tetrasole bromide (MTT reagent, Alfa Aesar, USA) in DMEM at final concentration 0.5 mg/mL and the cells were incubated for 4 h. The resulting formazan crystals were dissolved in DMSO, and the absorbance was measured at 540 nm with a Synergy MX plate reader (BioTeck, Winooski, VT, USA). The amount of formazan produced is assumed to be proportional to the number of living cells. Cell viability was measured as the ratio of the optical density of treated and untreated cells given in a percentage. The same procedure was applied for the estimation of dark toxicity of Pyr_4_CN_4_Pz, except that there was no exposure of cells to LED light. Since Pyr_4_CN_4_Pz, like the MTT dye, absorbs at a wavelength of 540 nm, we additionally evaluated the possible contribution of the optical density of porphyrazine to the results of the MTT assay (Appendix A.).Experiments were performed and repeated at least three times. Data analysis and calculation of IC_50_ were performed using the GraphPad Prism 6 software. 

### 3.8. Estimation of Singlet Oxygen Production by Pyr_4_CN4Pz

To establish the type of photoreactions caused by Pyr_4_CN_4_Pz, we measured the relative quantum yield of singlet oxygen generation using a chemical trap method. The solution of 200 μM 1,3-diphenylisobenzofuran (singlet oxygen trap, DPBF) and 5 μM Pyr_4_CN_4_Pz in dimethyl sulfoxide (DMSO) was irradiated with light at a dose of 0 to 100 J/cm^2^ using an LED light source (λ_ex_ 615–635 nm, 20 mW/cm^2^) at room temperature. The absorbance of DPBF at a wavelength of 420 nm was measured using a Synergy MX plate reader. The quantum yield of singlet oxygen generation was calculated relative to the quantum yield of the Photodithazine (0.56) from the ratios of the rate constants of DPIBF photobleaching (S).

### 3.9. Electrochemistry Experiment

The reduction potentials were determined using the cyclic voltammetry method (CV) using a three-electrode cell (potentiostat “Elins P-45X”) in an argon atmosphere. The working electrode was a glassy carbon (GC) electrode (d = 2 mm), and the auxiliary electrode was a platinum wire. A reference electrode was Ag/AgCl/KCl(sat.) with a waterproof diaphragm. The potential sweep rate was 0.1 V s^−1^. The solvent was acetonitrile. The background electrolyte is 0.1 M (NBu_4_)ClO_4_ («Aldrich»), twice recrystallized from aqueous EtOH and dried under a vacuum. The concentration of tricyanoethylenes was 2 mM. After CV experiments equimolar amount of ferrocene was added to the cell and the electrochemical experiments were repeated. The values of E_1/2_ are given versus the E_1/2_ of the Fc^+^/Fc couple.

#### 3.9.1. Irradiation of Solutions with Laser Diode at 405 nm

Irradiation of solutions was carried out using a 405 nm CW laser diode. Collimation lenses were used to expand the beam to 9 mm in diameter at a half-of-maximum intensity of the Gaussian profile at the cell point. During irradiation with a laser diode, the cell temperature was monitored using an Optris PI400 thermal imager. Laser diode power was measured with a Thorlabs power meter head S302C.

#### 3.9.2. Measurement of the Temporal Dynamics of Changes in Absorption

We used cells 10 mm wide and 0.5 mm or 1 mm path length made of quartz for transmittance measurements of PerTCNE and PyrTCNE solutions. A Shimadzu UV-1800 spectrophotometer was applied to obtain spectra. The spectral range was 200–1100 nm with a spectral resolution close to 1 nm.

#### 3.9.3. Setup for Two-Photon Absorption Measurement by Comparing the Fluorescence of Solutions

We used pulsed a Ti:Saphirre laser LOTIS LT2211 pumped by a master laser LOTIS LS2137 at a wavelength of 532 nm with 15 ns pulses at repetition rates of 1 to 10 Hz (Figure 11). Output laser beams were at wavelengths 400 or 800 nm. Mirror M was adjusted (moved to the corresponding fixed position) whenever the wavelength was changed. Two systems of apertures and collimation lenses were used to obtain the same configuration of beam profiles for two different wavelengths. The beam diameter for 400 nm radiation was 1.4 times smaller than that of the 800 nm beam due to the role of the second power of intensity for 800 nm radiation in the two-photon absorption process (under the assumption of the Gaussian beam profile). A Thorlabs power meter head S302C and A PM100D console were used for laser power measurements. A highly sensitive spectrometer Ocean Optics QE65Pro was used for spectral measurements (luminescence spectra). The spectral range of the spectrometer is 200 nm to 1000 nm. An Ocean Optics HL2000 calibration lamp was used in the spectrometer to ensure correct sensitivity in a wide spectral range. 

To verify the result of the measurement of the two-photon absorption cross-section the same experimental setup configuration was used for the determination of the two-photon absorption cross-section for Rhodamine B (Lambda Physik). Rhodamine B was dissolved in methanol with concentrations of 2 mmol/L (for experiments at wavelength 810 nm) and 0.2 mmol/L (for experiments at wavelength 405 nm). Applied pulse energy at 810 nm changes in the range of 2–30 mJ. Starting from the level of pulse energy 4–5 mJ bright luminescence appears (visible with the naked eye) and is registered by the spectrometer. Comparing and recalculating signals from photodetectors and spectrometer we find a two-photon absorption cross-section for Rhodamine B 1.5 × 10^-48^ cm^4^*s/photon with accuracy about 30%. [47].

## 4. Conclusions

To summarize, we developed new anti-Kasha rigidochromic pigments including new cyanoarylporphyrazine Pyr_4_CN_4_Pz with four peripheral pyren-2-yl groups. It exhibits a unique combination of properties of the therapeutic agent (effective photosensitizer for PDT) with double-sensory capacity allowing the real-time control of the therapeutic process simultaneously with the cell membrane viscosity and polarity changes. The obtained pigments demonstrate the potential efficacy of modulating successive photophysical intramolecular events inducing pronounced dual emission (DE). This unique form of anti-Kasha effect is found to be controlled by the changes of environmental polarity that can be utilized for intracellular membrane polarity sensing during PDT. Previously, we established that cyanoarylporphyrazines demonstrate excellent anchorage towards the intracellular membrane (mainly, endoplasmic reticulum and Golgi apparatus) and high brightness [11]. A potential advantage of our new compound compared to the recently reported high-performance plasma membrane polarity [48] is the ratiometric method for estimating local polarity (i.e., by the ratio of long-wavelength and short-wavelength emission intensities, which depends on local polarity). This considerably simplifies polarity monitoring, since there is no need to use complex and expensive equipment for measuring the sensor fluorescence lifetime, which depends on plasma membrane polarity. Thus, we showed that cyanoarylporphyrazines are potent immunogenic death inducers that could be effectively applied in the photodynamic therapy of cancer [49]. 

We synthesized new small molecule non-fullerene acceptors based on the aromatic fused-ring hydrocarbons substituted with tricyanoethylene fragments and studied their photophysical, electrochemical, and structural properties. We found that electron affinity increases with a number of fused aromatic rings. In addition, we found that the bandgap of PerTCNE is significantly narrower (1.62 eV) than that of widely studied PDI and its derivatives. Thus, PerTCNE can be attributed to the medium-bandgap materials which now attract growing interest for application in PV cells that harvest indoor light for Internet-of-Things (IoT) power electronic devices.

Moreover, the reported compounds can be promising as the potential dipolarity-enhanced photoinitiators for UV and two-photon 3D polymerization and microfabrication. PerTCNE showed a high two-photon absorption cross-section which is close to that of the widely used photoinitiators for two-photon 3D polymerization.

And finally, we believe that combination of sensory and therapeutic capacities of the new pigments is very promising for multifunctional applications ranging from photonics and biophotonics to biomedicine.

## Data Availability

The data used to support the findings in this study are available from the corresponding author upon request.

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
