# Peer review of "Novel Rigidochromic and Anti-Kasha Dual Emission Fluorophores Based on D-π-A Dyads as the Promising Materials for Potential Applications Ranging from Optoelectronics and Optical Sensing to Biophotonics and Medicine"

_ijms, 2023, doi:10.3390/ijms24065818_

Round 1

Reviewer 1 Report

ijms-2090890

Review of an article:

« Novel rigidochromic and anti-Kasha dual emission fluorophores based on D-π-A dyads as the promising materials for potential applications ranging from optoelectronics and optical sensing to biophotonics and medicine»

by Svetlana A. Lermontova, Maxim V. Arsenyev, Anton V. Cherkasov, Georgy K. Fukin, Andrey V. Afanasiev, Andrey V. Yudintsev, Ilya S. Grigoryev, Elena Yu. Ladilina, Tatyana Sergeevna Lyubova, Natalia Yu. Shilyagina, Irina V. Balalaeva, Larisa Grigorievna Klapshina, Alexandr V. Piskunov.

In International Journal of Molecular Sciences (ISSN 1422-0067).

Round 1

This work concerned new fluorescent materials exhibiting various sensory abilities. In this paper the new small fluorescent electron acceptors AntTCNE, PyrTCNE, and PerTCNE consisting of 3-5 fused aromatic rings substituted with tricyanoethylene fragments was synthesized. All the compounds reported exhibit pronounced rigidochromic properties, i.e. the strong sensitivity of their fluorescence to such the local stimuli as viscosity or rigidity.

The article was presented in a well-structured manner, with a good level of organization. Unfortunately, several statements within have weak evidence. Therefore, the referee suggested that the manuscript be improved with a major revision. The following is a list of specific concerns.

1.     The name of this work is too big. Please specify, and make it shorten.

2.     The authors do not cite any work from 2022.

To increase the quality of the introduction section, please, use the following references:

https://doi.org/10.1002/poc.4034;

https://doi.org/10.1016/j.jphotochem.2020.112932;

3.     Please provide water content data for the used solvent acetonitrile by Fisher titration or another method.

4.     Line 502. Fluorescence was studied in a stationary mode using a Perkin Elmer LS 55 502 spectrometer at 300–800 nm. Please, provide integration time at a single point, slits width, and detector voltage.

5.     Cell viability experiment.

-       Figure 7. Specify the dimension of the coordinate axis (Y) correctly.

6.     Using Levshin's law provides normalized spectra for Figure 8.

7.     Finally, it will be challenging to deduce helpful information on the different influences of solvent polarity and viscosity (section 2.5) solely based on photo stationary spectra. Is there any aggregation process for the aqueous solution?

8.     According to “E. Braslavsky Glossary of Terms Used in Photochemistry 3rd Edition, Pure Appl. Chem., 2007, 79, 293 —465”. The trivial reason for dual fluorescence is the coexistence of two independent fluorophores. Briefly describe primary fluorophore groups for your compounds.

9.     Specify the Conclusion section and provide possible implications of your study (including the significance of your study)—future research questions based on your findings.

Style guide issues

·       Line 436. Please, use the superscript character for 20 J/cm2 >>20 J/cm2.

·       Line 273: “well–known” should be spelled as well-known using a dash (-), not minus (–).

English spelling should be double-checked.

Redacted abstract is here below.

Reviewer 2 Report

The authors prepared small fluorescent electron acceptors AntTCNE, PyrTCNE and PerTCNE consisting of 3-5 fused aromatic rings substituted with tricyanoethylene fragments. Some validation experiments have been done to enrich the content of this work. However, this work is very routine and superficial, without enough novelty and supporting evidence to meet the high standard of INTERNATIONAL JOURNAL OF MOLECULAR SCIENCES. Thus, I strongly recommend this work to be rejected for publication. The other specific comments are as follows:

(1)   The design strategy of the probe design is not novel at all. What’s worse, the performance of the fluorophores is very weak, with rather short emission, weak fluorescence, poor response.

(2)   Manuscript writing is poor, full of format or grammar errors. For example, “macrocycle All the compounds” in line 20; “D-π-A (where π is a π-conjugating bridge, D is an aromatic π-donor, and A is a tricarbonitrile acceptor ” was repeatedly mentioned; “in the temperature range 270-300°Ð¡” in line 216; “that says Its intensity noticeably” in line 238; “abbey this rule” in line 276; “20 J/cm2 induced cell deat” in line 439; “laws (1 0 0 0 -1 0 0 0 -1) for AntTCNE and (-1 0 0 0 -1 0” in line 526; .

(3)   The NMR data expression should be checked..

Reviewer 3 Report

In this manuscript, the authors covalently linked tricyanoethylene group on anthracene, pyrene and perylene respectively to form a series donor-acceptor structures. A porphyrin derivative containing four cyanides was then synthesized by authors through conjugation of the pyrene derivative. Authors was then observed intramolecular charge polarization of these molecules under UV (~400 nm) irradiation. These compounds were claimed to be solvent viscosity sensitive. Moreover, authors observed two-photon absorption property for such molecules based on a self-developed calculation method. Lastly, the porphyrin derivative showed antimicrobial activity against a skin cancer cell (A-431) under red light irradiation (615–635 nm).

Revision is required for the manuscript by taking some arguments and suggestions into consideration, as follows.

1, Authors observed that fluorescent intensity of antTCNE is about 30 times higher than pyrTCNE and perTCNE under the same concentration in glycerin. Should the reason be discussed? Since antTCNE has the lowest extension coefficient at 400 nm than the other two compounds.

2, When demonstrating the fluorescence of the perTCNE in toluene, should the reason of using a different excitation light wavelength be discussed? Why was the same 400 nm light not used?

3, Authors observed that the fluorescence for these compounds is more intense in viscous glycerin than nonviscous water. However, without more comparison with other different solvents, the conclusion on viscosity sensitive property of the compounds should be conservative. For example, will protic and aprotic solvents be any different? Will nonpolar solvents with very different viscosity be any different? Will solubility of the compounds in different solvents affect the viscosity sensitive property?

4, Authors observed a very long electron relaxation time (214 h) through light transmission changes of these compounds. Should explanation and discussion be made on such phenomenon? Why molecular polarization of these compounds is so stable? Besides transmission, should time dependent UV-Vis absorption of the compounds be also investigated and discussed?

5, Instead of high-power laser z-scan technic, authors made calculations to determine the cross-section value of perTCNE. Should the same calculations to be made on similar known compounds such as perylene (δ = 0.7×10-48 cm4s/photon) to prove the accuracy of the calculation?

6, In order to illustrate the donor-acceptor property of the compounds and their potential application in photovoltaic field, should the cyclic voltammetry be carried out under the light irradiation condition? Should the dielectric property of the compounds be also investigated?

7, Authors observed that Pyr4CN4Pz has photo induced antimicrobial property against skin cancer cells. The mechanism of the photo killing should be investigated and discussed. What type of photochemistry the compound undergoes to kill the cell? Through Type I (electron transfer) or Type II (energy transfer) process? Reactive oxygen species determination and evaluation need to be investigated.

8, Cell viability was measured by MTT assay with the formation of formazan which was measured by plate reader at absorption of 540 nm. However, compounds themselves show absorption at 540 nm extending to over 700 nm. Will this lead to misleading reading of the viable cells?

9, Should the possibility of π-π stacking induced FL emission blue shift of Pyr4CN4Pz in the rigid environment be discussed as an alternative for hypothesized mechanism?

10, More structural characterizations such as C13 NMR, MASS and elemental analysis are needed for the synthesized compounds if no crystal structures are provided.

11, Some sentences need to be improved for better understanding, including typos in table S4 caption and line 454.

Round 2

Reviewer 1 Report

ijms-2090890

Review of an article: 

« Novel rigidochromic and anti-Kasha dual emission fluorophores based on D-π-A dyads as the promising materials for potential applications ranging from optoelectronics and optical sensing to biophotonics and medicine»

by Svetlana A. Lermontova, Maxim V. Arsenyev, Anton V. Cherkasov, Georgy K. Fukin, Andrey V. Afanasiev, Andrey V. Yudintsev, Ilya S. Grigoryev, Elena Yu. Ladilina, Tatyana Sergeevna Lyubova, Natalia Yu. Shilyagina, Irina V. Balalaeva, Larisa Grigorievna Klapshina, Alexandr V. Piskunov.

In International Journal of Molecular Sciences (ISSN 1422-0067).

Round 2

All concerns have been addressed properly, thus I recommend the publication.

Author Response

We agree with this comment.

Reviewer 2 Report

Through modification, the language of the article has been improved; However, the innovation of the article is still low. Rejection is suggested .

Author Response

We do not agree with this comment.

Reviewer 3 Report

Author’s response to the comments is very much appreciated. However, some of the responses and the revised manuscript itself did not resolve all the issues and questions in the original comments. Without resolving the existing issues, the manuscript is not appropriate to be published in IJMS. Please see attachments for details.

Round 3

Reviewer 3 Report

The revised manuscript and author's response have resolved all the questions and issues. Details are enclosed in attachment. 

Author Response

The reviewer considers that the revised manuscript and author’s response have resolved all the questions and issues. We agree with this comment. No revision is needed.